# Novel Long Noncoding RNA miR205HG Functions as an Esophageal Tumor-Suppressive Hedgehog Inhibitor

**DOI:** 10.3390/cancers13071707

**Published:** 2021-04-03

**Authors:** Jee Hoon Song, Alan H. Tieu, Yulan Cheng, Ke Ma, Venkata S. Akshintala, Cem Simsek, Vishnu Prasath, Eun Ji Shin, Saowanee Ngamruengphong, Mouen A. Khashab, John M. Abraham, Stephen J. Meltzer

**Affiliations:** 1Division of Gastroenterology and Hepatology, Department of Medicine, Johns Hopkins School of Medicine, Baltimore, MD 21287, USA; jsong37@jhmi.edu (J.H.S.); alan.tieu@jhmi.edu (A.H.T.); yulancheng@jhmi.edu (Y.C.); kma5@jhmi.edu (K.M.); vakshin1@jhmi.edu (V.S.A.); csimsek2@jhmi.edu (C.S.); vprasat1@jhu.edu (V.P.); eshin3@jhmi.edu (E.J.S.); sngamru1@jhmi.edu (S.N.); mkhasha1@jhmi.edu (M.A.K.); jabrah14@jhmi.edu (J.M.A.); 2Department of Medicine, University of Maryland School of Medicine, Baltimore, MD 21287, USA; 3Division of Gastroenterology and Hepatology, Department of Medicine, Eastern Virginia Medical School, Norfolk, VA 23456, USA

**Keywords:** esophageal adenocarcinoma (EAC), Barrett’s esophagus, long noncoding RNA (lncRNAs), hedgehog inhibitor

## Abstract

**Simple Summary:**

Barrett’s esophagus (BE) is a precursor to esophageal adenocarcinoma (EAC). Long noncoding RNAs (lncRNAs) have been identified as key regulators of biological pathways and we identified lncRNA, *miR205HG,* as a tumor suppressor in the development of Barrett’s esophagus and esophageal adenocarcinoma, in part through its effect on the Hedgehog signaling pathway. The aims of the current study were: (1) to study involvement of *miR205HG* in the development of BE and EAC (2) to clarify the role of *miR205HG* in in vitro and in vivo experiments; and (3) to investigate the mechanism of *miR205HG* involving the Hedgehog (Hh) signaling pathway

**Abstract:**

Barrett’s esophagus (BE) is a precursor to esophageal adenocarcinoma (EAC). Recently, long noncoding RNAs (lncRNAs) have been identified as key regulators of biological pathways. However, involvement of lncRNAs in the development of BE and EAC has not been well-studied. The aims of the current study were: (1) to study involvement of the lncRNA, *miR205HG*, in the development of BE and EAC; (2) to clarify the role of *miR205HG* in in vitro and in vivo experiments; and (3) to investigate the mechanism of *miR205HG* involving the Hedgehog (Hh) signaling pathway. These experiments revealed that *miR205HG* was downregulated in EAC vs. normal esophageal epithelia (NE) as well as in EAC cell lines, and its forced overexpression inhibited EAC cell proliferation and cell cycle progression in vitro. Similarly, overexpression of *miR205HG* inhibited xenograft tumor growth in mice In vivo. Finally, we show that one mechanism of action of *miR205HG* involves the Hh signaling pathway: *miR205HG* and Hh expression levels were inversely correlated in both EAC (r = −0.73) and BE (r = −0.83) tissues, and in vitro studies revealed details of Hh signaling inhibition induced by *miR205HG.* In conclusion, these findings establish that lncRNA *miR205HG* functions as a tumor suppressor in the development of BE and EAC, at least in part through its effect on the Hh signaling pathway.

## 1. Introduction

Esophageal adenocarcinoma (EAC) is the predominant histopathologic subtype of esophageal cancer in the United States and other Western countries [1]. EAC incidence has increased sevenfold over the past three decades, while that of other common cancers has declined [1,2]. Most EACs are detected at advanced stages, with five-year survival rates < 20% [3]. The rising incidence and poor prognosis of this cancer emphasize the need for understanding key etiologic factors. Barrett’s esophagus (BE) is the major predisposing factor to EAC [4]. A diagnosis of BE confers an 11- to 30-fold increase in EAC risk, while BE affects 1.6–6.8 percent of the general population [5]. Although endoscopic surveillance for BE may improve EAC-associated mortality [6], surveillance BE patients represent only a fraction of total EAC patients [5]. Thus, it is crucial to improve our understanding of EAC’s underlying molecular mechanisms. A better understanding of the molecular basis of BE and EAC may yield earlier detection, more effective individualized cancer risk evaluation, and novel therapeutic approaches.

Recently, long noncoding RNAs (lncRNAs) have added a new layer of complexity to the molecular lexicon, emerging as key regulators of diverse biological pathways. These studies show that lncRNAs contribute to many cellular processes, including protein-coding gene regulation, genomic imprinting, mRNA processing, and cell differentiation and development [7]. Dysregulated lncRNAs have been implicated in human carcinogenesis, including HOTAIR in breast cancer, MALAT-1 in lung cancer, HULC and HEIH in hepatocellular carcinoma, SPRY4-IT1 in melanoma, and PCGEM1 and ANRIL in prostate cancer [8,9,10,11,12,13,14]. We recently identified two specific lncRNAs (HNF1A-AS1 and AFAP1-AS1) as being involved in BE-associated neoplastic progression [15,16].

Aberrant activity of embryologic signaling pathways has been implicated in the development of BE and EAC. Specifically, perturbations of the Hedgehog (Hh), bone morphogenetic protein (BMP), wingless-type MMTV integration site family (WNT) and retinoic acid (RA) signaling pathways have been reported in both BE and EAC [17]. Among these pathways, Hh signaling stands out for its importance in embryonic development of the gastrointestinal epithelium, including the esophageal epithelium, as well as its role in intestinal epithelial homeostasis. The Hh pathway was first identified in *Drosophila* as an important regulator of proper embryonic patterning and is highly evolutionarily conserved [18]. Moreover, Hh signaling connects to other downstream signaling pathways, including BMP signaling. Further studies have revealed key roles for Hh signaling in embryonic development, cell proliferation, tissue polarity, and carcinogenesis [19,20,21,22,23]. Sonic Hedgehog (SHH), the best-studied ligand in the Hh signaling pathway, binds to the membrane receptor Patched (PTCH) [24,25]. This binding relieves inhibitory actions of PTCH on Smoothened (SMO), a G-protein coupled receptor [18]. The activated SMO protein subsequently activates the glioma-associated oncogene (Gli) transcription factor, triggering further downstream signaling cascades [26]. Although SHH-signaling is not active in normal adult esophageal cells, acid and bile reflux trigger abnormal activation of SHH-signaling in the esophagus, thereby contributing to BE pathogenesis [27]. In further support of this mechanism, SHH-signaling was found to contribute to BE development by activating BMP signaling and inducing epithelial expression of SOX9, a transcription factor associated with intestinal stem cells [28]. Upregulated SHH-signaling is often observed in frank human EAC, and this enhanced signaling appears to stimulate EAC cell survival and proliferation in vitro [27,28,29]. In summary, the SHH-signaling pathway appears to trigger both the origin of BE and its progression toward dysplasia and EAC, thus providing an attractive target for the prevention of both premalignant BE and full-blown esophageal malignancy.

The involvement of lncRNAs in the origin and progression of BE and EAC are not well-understood. In the current study, we sought to (1) establish the key lncRNA, *miR205HG*, in development of EAC and BE; (2) clarify its BE- and EAC-suppressive role in vitro and in vivo; and (3) investigate the mechanism of action *miR205HG* via Hedgehog (Hh) signaling. The 5909 bp miR-205HG (mir-205 host gene; also called LINC00510, NONHSAG004163.1, NPC-A-5) comprises five exons and exists as several splice variants (Figure 1); it is located at chromosome location 1q32.2 (209,427,530-209,433,438, plus strand; GRCh38/hg38) and encodes also the human miR-205 gene which gives rise to two mature miRNAs, miR-205-5p and miR-205-3p. The miR-205 family is frequently silenced in advanced cancer.

## 2. Materials and Methods

### 2.1. Identification of a Novel Long Noncoding RNA Downregulated in Barrett’s Esophagus and Esophageal Adenocarcinoma

#### 2.1.1. Cell Lines and Tissues

Primary nonimmortalized esophageal epithelial cells (HEEpiC), along with the EAC cell lines FLO-1 (RRID:CVCL_2045), SKGT4 (RRID:CVCL_2195), and OE33 (RRID:CVCL_0471), were purchased from ScienCell Research Laboratories (Carlsbad, CA, USA), Sigma Chemical (St. Louis, MO, USA), and the European Cell Culture Collection (Porton Down, UK), respectively. The BE cell lines CP-A (RRID:CVCL_C451; also called QHTRT) and CP-C (RRID:CVCL_C453; also called GiHTRT) were gifts of Dr. Peter Rabinovitch (University of Washington [30,31] and they are not commercial available). However, there are published articles using these cell lines [32]. The JH-EsoAd1 (RRID:CVCL_8098) EAC cells were provided by Dr. James R. Eshleman (Johns Hopkins University). Normal primary esophageal epithelial cells and Barrett’s cells were cultured in EpiCM-2 medium supplemented with epithelial cell growth supplement-2 and 0.5% penicillin/streptomycin at 37 °C. All other cell lines were cultured in MEM medium supplemented with 10% fetal bovine serum (Invitrogen, San Diego, CA, USA), unless otherwise stated. All human cell lines have been authenticated using STR (short tandem repeat) profiling within the last three years. All experiments were performed with mycoplasma-free cells. Human tissues were obtained at endoscopies performed for clinical diagnostic indications and stored in liquid nitrogen prior to RNA extraction. All patients provided written informed consent under IRB-approved protocols (#MO16M37 and #NA-00004336 approved on 2/19/2106 and renewed on 10/20/2020) at JHU, University of Maryland, or the Baltimore Veterans Affairs Medical Center. Informed consent was obtained from all subjects involved in the study. All tissues were histopathologically confirmed as NE, BE, or EAC. Two matched sets of NE-BE-EAC tissues and two matched pairs of NE-BE tissues were studied by RNAseq. In addition, twenty-nine matched sets of NE-EAC tissues and fourteen matched pairs of NE-BE tissues from patients without EAC were analyzed for differential expression of *miR205HG*.

#### 2.1.2. RNA Extraction, miRNA Extraction, and Quantitative Reverse Transcription Polymerase Chain Reaction (qRT-PCR)

Total RNA was extracted using Trizol reagent (Invitrogen, Carlsbad, CA, USA) and RNeasy kits (Qiagen, Valencia, CA, USA), combined with RNase-free DNase (Qiagen, Germantown, MD, USA). First-strand complementary DNA (cDNA) was synthesized from 500 ng of total RNA using RevertAid™ cDNA synthesis kits (Fermentas, Glen Burnie, MD, USA). qRT-PCR was performed using iQ SYBR Green Supermix on an iQ5 Multicolor Real-Time PCR Detection System (Bio-Rad, Hercules, CA, USA). Sequences of primers (Integrated DNA Technologies, Coralville, IA, USA) are shown in Table 1. Total RNAs from normal tissues of 20 different organs were purchased from Applied Biosystems (Foster City, CA, USA; Human total RNA survey panel kit). RNAs were stored at −80 °C before and after analysis. cDNA was synthesized from 10 ng of total RNA using TaqMan Reverse Transcription Kits (Applied Biosystems) and a specific primer from Taqman MicroRNA Assays (Applied Biosystems, Waltham, MA, USA). qRT-PCR was performed using iQ Supermix (Bio-rad, Hercules, CA, USA) and a labeled probe from Taqman MicroRNA Assays (Applied Biosystems, Waltham, MA, USA). RNU6B small nuclear RNA served as an internal normalization control and compared using the delta-delta Ct method. Two independent experiments were performed, each of which was carried out in triplicate.

#### 2.1.3. Next-Generation RNA Sequencing

RNA was checked for quality using a BioAnalyzer (Agilent, Santa Clara, CA, USA) then depleted of rRNA (ribosomal ribonucleic acid) using an Epicentre RiboZero kit (Illumina, San Diego, CA, USA). Remaining RNA (≤5%) was sheared using Covaris system, then prepared as directional RNAseq libraries using dUTP/UNG system. Paired-end 100 bp sequencing using Illumina HiSeq 2000 (San Diego, CA, USA) was performed for each sample. Initial data processing was done via Illumina Sequence Control Software (SCS) (Illumina, San Diego, CA, USA) and Pipeline 1.n software (Illumina, San Diego, CA, USA), which perform functions of image analysis (Firecrest), base-calling (Bustard) and alignment of sequence tags to the appropriate reference genome (ELAND), sorting samples by index sequences (up to four per lane). For transcriptional profiling, to determine counts of sequences from each gene relative to others, we normalized counts to total reads to allow intersample comparisons.

### 2.2. miR205HG Function in Esophageal Tumorigenesis

#### 2.2.1. Generation of *miR205HG*-Stable Cell Lines and Transfection of *miR205HG* Plasmid and miRNA Mimics

An *miR205HG* insert was cloned into pcDNA3.1(-) (Invitrogen, Carlsbad, CA, USA). Cells were seeded onto six-well plates at 1.5 × 10^5 cells/well. After 24 h, 1.5 μg of pcDNA3.1(-) either lacking or containing an *miR205HG* insert were transfected using BioT transfection reagent (Bioland Scientific LLC, Paramount, CA, USA), according to the manufacturer’s protocol. RNA was harvested after 48 h of transfection. For generation of *miR205HG*-stable cell lines, cells underwent three weeks of selection in 600 μg/mL G418, and several monoclonal colonies were selected per cell line and expanded. *miR205HG* expression levels were confirmed by qRT-PCR for each stable clone. Synthesized RNA duplexes of miRNA mimics were purchased from Dharmacon (Lafayette, CO, USA). 50~70% confluent FLO-1 EAC cells were transfected with 50–60 nM of negative control mimic or miR-205 mimic using Lipofectamine RNAiMAX (Invitrogen, Carlsbad, CA, USA). RNA was harvested after 48 h of transfection.

#### 2.2.2. Cell Proliferation Assays 

SKGT4 and FLO-1 *miR205HG* stably-transfected cells were reseeded onto 96-well plates at 1000 cells/well (Day 0). Cell proliferation was assessed at Days 1, 3, and 5, using cell proliferation reagent water-soluble tetrazolium salt (WST-1) (Roche, Mannheim, Germany). Ten microliters of reagent were added to each well, incubated at 37 °C for 1 h, and optical density measured at 660 nm (background) and 440 nm (signal) using a plate reader (Molecular Devices, Sunnyvale, CA, USA). Independent experiments were repeated three times, with six replicates per experiment.

#### 2.2.3. Cell Cycle Analyses by Flow Cytometry

Flow cytometric analysis of DNA content was performed to assess cell cycle phase distribution. SKGT4 and FLO-1 *miR205HG* stable clones were incubated with PI staining buffer (PBS 0.1 mg/mL PI, 0.6% NP40, 2 mg/mL RNase A) for 30 min on ice (Roche Diagnostics, IN, USA). DNA content was determined using FACSCalibur (BD Biosciences, San Jose, CA, USA) and Cell Quest software (BD Biosciences, San Jose, CA, USA) for histogram analysis.

#### 2.2.4. Clonogenic Assays

SKGT4 and FLO-1 *miR205HG* stably transfected cells were trypsinized into a single-cell suspension. One hundred cells were plated per well of a six-well plate and maintained for 14 days to allow colony formation. Clones containing > 50 cells were counted using a grid. Three independent experiments were performed.

#### 2.2.5. Invasion Assays

Cell suspension in serum-free media was added into sterile 8 µm pore basement membrane matrix-coated inserts (Cell Biolabs, San Diego, CA, USA). Inserts were lowered into 24-well plates containing 20% serum- containing media and incubated at 37 °C for 24 h. Invasive cells were dissociated and subsequently detected by CyQuant GR Fluorescent Dye (Invitrogen, Carlsbad, CA, USA).

#### 2.2.6. Injection of Tumor Cells into Nude Mice

All animal studies were approved by the JHU Animal Care Committee and conducted in accordance with IACUC policy. Fifteen female 9–10 weeks old nude mice were obtained (Charles River, Boston) and divided into three groups (5 mice per group). Their average weights were 28.5 ± 0.3 g. FLO-1 *miR205HG*-stable cells were used to establish tumor xenografts. Control mice were injected with FLO-1 empty vector-only cells; two treatment groups were injected with two distinct FLO-1 *miR205HG*-stable clones. Each mouse was subcutaneously injected in left or right rear flank with 0.2 ml of tumor cells (2 × 10^6^ cells) mixed in Matrigel. After two weeks, mouse tumor volume was measured three times weekly. Tumor width (W) and length (L) were measured using digital calipers, and volume determined using formula V = ½(L × W^2^). Before injecting tumor cells, we anesthetize mice using of 2–4% isoflurance. Mice were euthanized using CO_2_. 

### 2.3. Identifying miR205HG’s Function as an Esophageal Tumor-Suppressive Hedgehog Inhibitor 

#### 2.3.1. Transient Transfection of *miR205HG* Plasmid

A *miR205HG* insert was cloned into pcDNA3.1(-) (Invitrogen, Carlsbad, CA, USA). CP-C and CP-A BE cells were seeded onto six-well plates at 1.5 × 10^5^ cells/well. After 24 h, 1~1.5 μg of pcDNA3.1(-) either lacking an insert or containing a *miR205HG* insert was transfected using BioT transfection reagent (Bioland Scientific LLC, Paramount, CA, USA), according to the manufacturer’s protocol. GiHTRT cells were more sensitive to transfection reagent than CP-A cells, so 1.0 ug of *miR205HG* plasmid was used for CP-C cells while 1.5 μg of *miR205HG* plasmid was used for CP-A cells. RNA was harvested after 48 h of transfection.

#### 2.3.2. Transfection of miR Mimics

Synthesized RNA duplexes of miR mimics were purchased from Dharmacon. 50~70% confluent FLO-1 EAC cells were transfected with 50–60 nM of mimic negative control or miR-205 mimic using Lipofectamine RNAiMAX. RNA was harvested after 48 h of transfection.

#### 2.3.3. miRNA Binding Site Prediction

Three online miRNA target prediction engines (miRanda, Diana Tools 5.0: microT-CDS, and RNAhybrid 2.2- Athena Innovation, Volos, Greece) were used to predict miR-205 binding sites within *PTCH1* the three prime untranslated region (3′UTR). Since no consensus was found among all three engines, each prediction result was considered equally important.

#### 2.3.4. Transfection and Reporter Assays

Confluent cultures of *miR205HG*-stable cells were plated onto six-well plates at 1.5 × 10^5^ cells/well. On the next day, we cotransfected cells with Renilla luciferase and either pmirGLO-PTCH1 or pGL3-8xGLI (gifted by Dr. Philip Beachy, Stanford University, Stanford, CA, USA; the 8X Gli fragment was subcloned into the pGL3 promoter vector) using BioT transfection reagent (2 μL well) according to manufacturer’s protocol. After cells reached saturation density (one–two days), they were changed to low-serum medium (0.5% bovine calf serum). Cells were lysed and transferred to 96-well plates prior to luciferase reporter assays using Dual-Glo luciferase assay kits (Promega, Madison, WI, USA). Luminescence intensity was measured by VICTOR2 fluorometry (Perkin Elmer, Waltham, MA, USA), and luminescence intensity of Firefly luciferase was normalized to that of Renilla luciferase.

#### 2.3.5. Three Prime Untranslated Region (3′UTR) *PTCH1* Plasmid Construction

The full-length PTCH1 3′ untranslated region (3′UTR), containing several predicted miR-205 binding sites, was amplified from genomic DNA using linker primers containing *NheI* and *XhoI* restriction sites. Amplicons were cut and cloned into pmirGLO-dual luciferase vector just downstream of firefly luciferase structural gene (Promega). Similarly, three truncated PTCH1 3′UTRs were constructed and cloned into pmiRGLO-dual luciferase vector. Empty vector lacking PTCH1 3′UTR was used as negative control.

#### 2.3.6. Western Blotting

Cell lysates were separated by sodium dodecyl sulfate (SDS)–polyacrylamide gel electrophoresis (Bio-Rad, Hercules, CA, USA). After electrophoresis, protein was transferred onto polyvinylidene fluoride (PVDF) membranes (Millipore, Bedford, MA, USA), immunoblotted with tris buffered saline (TBS) containing 5% nonfat dry milk, washed with 0.1% TBST, and probed with 1:1000 anti-SHH rabbit monoclonal antibody (#8358S; Cell Signaling, Danvers, MA, USA) and 1:7000 antihuman β-actin rabbit monoclonal antibody (#A5060; Sigma-Aldrich, Bedford, MA, USA). Horseradish peroxidase-conjugated antirabbit goat IgG (1:3000) (#401393, Calbiochem, San Diego, CA, USA) and ECL Western blotting detection kits (Amersham Pharmacia Biotech, Piscataway, NJ, USA) were used for target protein visualization.

#### 2.3.7. Chromatin Immunoprecipitation Assay (ChIP)

FLO-1-pcDNA 3.1(-) (control) and FLO-1-pcDNA 3.1-*miR205HG* were used for immunoprecipitation with RNA ChIP-IT kits (Active Motif, Carlsbad, CA, USA) according to manufacturer’s protocol. Normal rabbit IgG (control), Suz12 (#3737; Cell Signaling), and Ezh2 (#5246; Cell Signaling) antibodies were used for immunoprecipitation. Primers targeting SHH promoter region amplified a product from sheared immunoprecipitated chromatin template. Nontargeting primers were used as negative control for primer specificity (Appendix A).

### 2.4. Statistical Analysis

Statistical analysis was performed using GraphPad Prism 6.04 (GraphPad, La Jolla, CA, USA). Experimental results were evaluated using two-tailed Student’s *t*-test or Spearman rank correlation test. All values were expressed as mean ± SD. Statistical significance was noted at *p*-value < 0.05, and three independent triplicate experiments were performed for cell biological assays, unless otherwise stated.

### 2.5. Data Availability

The data that support the findings of this study are available from the corresponding author upon reasonable request. RNA-seq data of the two NE-BE-EAC matched tissue sets and the two NE-BE matched tissue pairs are available at GEO (Accession Number GSE48240).

## 3. Results

### 3.1. Identification of miR205HG and Its Downregulation in Barrett’s Esophagus (BE) and Esophageal Adenocarcinoma (EAC) Cell Lines and Tissues

To identify tumor suppressor lncRNAs involved in Barrettogenesis and esophageal adenocarcinogenesis, we performed RNA-seq of two NE-BE-EAC matched tissue sets and two NE-BE matched tissue pairs (full data available at GEO, Accession Number GSE48240). This RNA-seq analysis identified 1531 lncRNAs. Among these 1531 lncRNAs, we prioritized lncRNAs with highest expression in NE (NE normalized copy number ≥ 10) that were sequentially downregulated during the NE-BE-EAC progression continuum (cut-off fold-change ≥ 1.5). This filtering process identified 11 lncRNAs (Table 2), among which CTA-55I10.1 (subsequently renamed miR205HG) had the highest NE/BE and NE/EAC fold changes of these 11 candidate lncRNAs. *miR205HG* is a microRNA (miRNA)-host gene, which harbors within its intron 3-exon 4 junction a miRNA-containing hairpin that serves as the template for two distinct miRNAs, miR-205 and miR-205* (Figure 1). The transcript splice variant 004 was chosen because it showed the most robust expression out of all variants.

After finding differential expression of *miR205HG* in our original RNAseq data, we proceeded to validate its differential expression in esophageal cells and primary human BE and EAC tissues. We performed quantitative real-time polymerase chain reaction (qRT-PCR) in six esophageal cell lines (two BE-derived and four EAC-derived), as well as in paired BE and EAC tissue samples (14 NE-BE pairs + 29 NE-EAC pairs). Results showed that *miR205HG* was either undetectable or substantially and significantly downregulated in both BE-derived cell lines (CP-C and CP-A) and 4/4 EAC-derived cell lines (FLO-1, JH-EsoAd1, OE33, and SKGT4) vs. primary normal esophageal epithelial cells (HEEPiC) (*p*-values < 0.001; Figure 2). Northern blotting of *miR205HG* confirmed these qRT-PCR results (Figure 3). Subsequently, 29 matched NE-EAC tissue pairs were assessed for *miR205HG* expression; 26/29 EACs (89.7%) showed *miR205HG* downregulation vs. matched NE tissues (*p*-value < 0.0001; Figure 4). Similarly, 14 NE-BE tissue pairs from patients without dysplasia were tested; 14/14 BE tissues (100%) demonstrated significant *miR205HG* downregulation vs. paired NEs (*p*-value < 0.0001; Figure 5). Not only was *miR205HG* significantly downregulated in most EAC cells, but it was also significantly reduced in BE cells. These findings strongly support the function of *miR205HG* as a BE- and EAC-suppressive gene, with *miR205HG* downregulation driving both early preneoplastic change (from NE into BE) and later neoplastic progression (from BE into EAC). Finally, in a survey of multiple normal human tissues, we found that in addition to being highly expressed in normal esophageal cells and tissues, *miR205HG* was expressed abundantly in normal cervix, prostate, trachea, and thymus (Figure 6). Thus, these data suggest that *miR205HG* exerts important biological functions not only in the esophagus, but also other organs, and its downregulation may contribute to diseases of these other organs, such as prostate cancer.

### 3.2. Part II

#### 3.2.1. Overexpression of *miR205HG* Inhibits EAC Cell Proliferation, Colony Formation, Invasion/Migration, Induces EAC G0-G1 Cell Cycle Arrest 

To determine functional consequences of increased *miR205HG* expression in EAC cells, several in vitro assays were performed in Flo-1 and SKGT4 cells. We designed a construct (*miR205HG* vector) lacking its wild-type intronic miR-205 hairpin structure. This design enabled us to focus on effects of the overexpressed lncRNA per se, without any interference from the competing transcript miR-205. This construct was used to generate independent stably transfected clones of two EAC cell lines, SKGT4 and FLO-1; qRT-PCR experiments confirmed *miR205HG* overexpression in both of these clonal populations (Figure 7 Notably, miR-205 was undetectable in both clones (data not shown). *miR205HG* expression levels in these two clones were not as high as those in normal esophageal cells, but significantly higher than in empty vector-transfected clones. Next, *miR205HG*-stable clones were used in WST-1 assays to detect effects of *miR205HG* overexpression on proliferation of SKGT4 and FLO-1 EAC cells vs. vector-only transfected negative control, forced overexpression of *miR205HG* decreased cell proliferation on Day 5 by 45% and 69% in SKGT4 and FLO-1 cells, respectively (*p*-values all < 0.001; Figure 8). Consistent with these cell proliferation results, colony size was also significantly smaller and relative colony number was reduced in FLO-1 and SKGT4 *miR205HG*-stable clones vs. control cells. Similarly, clonogenic assays revealed that *miR205HG* overexpression caused substantial reductions in colony formation in SKGT4 (49% reduction; *p*-value < 0.02) and FLO-1 (69% reduction; *p*-value < 0.01) cells (Figure 9). These findings suggest that *miR205HG* suppresses the ability of EAC to proliferate and undergo “unlimited” cell division, two well-characterized malignant processes.

We also performed invasion/migration assays on stable FLO-1 clones and transient *miR205HG*-overexpressing OE33 cells using Matrigel transwell chambers (because SKGT4 cells did not attach to invasion wells). *miR205HG* overexpression caused substantial decreases in invasion in FLO-1 (58% reduction; *p*-value < 0.001) and OE33 (39% reduction; *p*-value < 0.03) EAC cells (Appendix A). Subsequently, to delineate potential mechanisms underlying the growth-inhibitory effects of *miR205HG* overexpression, we assessed cell cycle progression in *miR205HG*-stably transfected FLO-1 cells. Flow cytometric cell cycle assays demonstrated that vs. empty vector-transfected control cells, *miR205HG* overexpression caused significant accumulation of cells at G0/G1-phase and decrease at S-phase (Figure 10). Proportions of apoptotic cells in *miR205HG*-overexpressing vs. empty-vector negative control cell populations were similar (data not shown). Thus, *miR205HG*-mediated inhibition of EAC cell proliferation and colony formation are mediated by modulation of the G1-S checkpoint, rather than by apoptosis. Since *miR205HG*-stable clones did not manufacture any miR-205, these results establish that this host gene lncRNA per se independently exerts tumor-suppressive effects without any tumor-suppressive activity of its cognate miRNA transcript.

#### 3.2.2. *miR205HG* Inhibits in vivo EAC Tumor Growth 

Having shown that *miR205HG* inhibits EAC cell growth in vitro, we next sought to demonstrate that it reduces esophageal tumor growth in vivo. We injected mice with FLO-1 cells stably transfected with vectors either lacking or containing a *miR205HG* insert. Fifteen athymic nude mice were evenly divided into three groups: group 1 mice were injected with cells lacking miR205HG insert (control group), while groups 2 and 3 received cells containing *miR205HG* insert (treatment groups 1 and 2). Treatment groups 1 and 2 were injected with two distinct clonal populations of *miR205HG* insert-containing cells. Interestingly, tumors in *miR205HG*-treated animals were already significantly smaller than in untreated controls on Day 14 (control vs. treatment group 1, *p*-value = 0.003; control vs. treatment group 2, *p*-value = 0.004); moreover, mean tumor volumes in treated groups 1 and 2 (37.1 ± 25.6 mm^3 and 40.5 ± 20.5 mm^3, respectively) were significantly smaller than in control group (83.4 ± 27.4 mm^3) (Figure 11). Significant decreases in tumor growth were observed in treated animals up to Day 36 (control vs. treatment group 1, *p*-value = 0.031; control vs. treatment group 2, *p*-value = 0.017); mean tumor volumes in treated groups 1 and 2 (140.9 ± 67.5 mm^3 and 131.9 ± 44.1 mm^3, respectively) remained significantly smaller than in the control group (231.0 ± 54.0 mm^3) (Figure 11). 

Next, we excised and stained xenografts with hematoxylin-eosin for histological analysis. Interestingly, while xenografts derived from vector-only control EAC cells contained densely packed sheets of cells with only occasional individual cell necrosis and leukocyte infiltration, xenografts derived from *miR205HG*-treated EAC cells contained areas of hemorrhage and necrosis (arrows), leukocyte infiltration (miR205HG #1), tumor cell loss, accumulation of proteinaceous fluid, leukocyte infiltration, and necrotic tumor cells (arrowheads) (miR205HG #2) (Figure 12). In summary, mean tumor size in treatment groups was significantly smaller than in control group throughout these experiments (Figure 13). This finding suggests that overexpression of *miR205HG* inhibits in vivo EAC growth. In addition to reducing tumor growth, forced *miR205HG* overexpression disrupted and killed tumor cells. These results support further investigation to develop novel therapeutic regimens leveraging *miR205HG* in the treatment of EAC.

### 3.3. Identifying miR205HG’s Function as an Esophageal Tumor-Suppressive Hedgehog Inhibitor 

#### 3.3.1. Key Sonic Hedgehog (SHH)-signaling Genes Are Upregulated in BE and EAC Cell Lines, and *miR205HG* and SHH Expression Levels Are Inversely Correlated in BE- and EAC-Matched Tissues

Knowing the functional importance of the Hh (Hedgehog) pathway in regulation of cell proliferation, invasion, colony formation, and cell cycle [33], we reasoned that this pathway was intimately involved in BE and EAC development, and that *miR205HG* could act by perturbing this pathway. We began by measuring expression levels of SHH pathway genes in BE and EAC cell lines vs. primary NE (HEEpiC) cells. These experiments revealed that, vs. NE cells, RNA levels of key SHH pathway genes *SHH, PTCH1, SMO*, and *Gli1* were upregulated in all BE cell lines and EAC cell lines examined (*p*-values < 0.005) (Figure 14). Interestingly, among four SHH pathway genes studied, average expression fold-change in BE and EAC cells was greatest for SHH and Gli1. These results agree with previous studies reporting upregulated expression of SHH-signaling genes in BE and EAC [27,28]. Matched NE-EAC tissue pairs from 26 EAC patients already been tested for *miR205HG* expression were then tested for SHH expression by qRT-PCR. SHH expression was upregulated vs. NE in 23/26 EACs studied (*p*-value < 0.0001) (Figure 15). Similarly, in matched NE-BE tissue pairs, SHH was also upregulated in BE relative to NE in 14/14 cases studied (*p*-value < 0.0001) (Figure 16). Interestingly, in matched BE and EAC tissues, levels of *miR205HG* and SHH were inversely correlated (r = −0.73, *p*-value = 0.0001; r = −0.83, *p*-value = 0.0004, respectively) (Figure 16). This strong inverse correlation between miR205HG and SHH is encouraging, since SHH upregulation and *miR205HG* downregulation are both implicated in genesis and progression of BE and EAC. We speculated that *miR205HG* could directly or indirectly inhibit SHH expression in NE cells. We therefore utilized *miR205HG* insert-containing plasmids to test this hypothesis.

#### 3.3.2. SHH Downregulation in *miR205HG*-Transfected EAC Clones

*miR205HG*-stable EAC clones were used to assess the effect, if any, of forced miR205HG overexpression on key SHH-signaling genes. Relative to vector-only stably transfected negative control EAC cell clones, *miR205HG*-transfected clones showed significant reductions in Shh, SMO, and Gli1 expression (Figure 17). SHH protein levels were also markedly reduced in miR205HG-containing SKGT4 and FLO-1 cell clones by Western blotting (Figure 18). SMO and Gli1 protein levels did not change with forced *miR205HG* overexpression (data not shown). The lack of change in these protein levels could have resulted from a negligible impact of *miR205HG* on protein translation of further downstream Hh pathway targets. Nevertheless, these experiments establish that *miR205HG* inhibits SHH by reducing its RNA and protein levels. Thus, we hypothesized that *miR205HG*’s inhibition of Hh disrupted downstream Hh signaling.

#### 3.3.3. miR-205 Directly Binds to PTCH1 3′UTR 

We had previously considered the possibility that miR-205 could act synergistically with its own host gene, *miR205HG*. We therefore evaluated synergistic effects of these two cognate transcripts. Specifically, after establishing that miR205HG inhibits SHH-signaling, we examined whether miR-205 could also suppress this pathway. Again, we measured expression of the SHH pathway genes *SHH, PTCH1, SMO*, and *Gli1* after transiently transfecting *FLO-1* EAC cells with either negative control mimic or miR-205 mimic. Among these four genes, only PTCH1 showed significant (*p*-value = 0.0001) reduction when transfected with 60 nM of miR-205 mimic (Figure 19). We then speculated that miR-205 directly targeted the PTCH1 mRNA. To assess direct binding of miR-205 to the PTCH1 3′UTR, we constructed luciferase vectors. Three online miRNA target prediction engines (miRanda, Diana Tools 5.0: microT-CDS, and RNAhybrid 2.2) were used to predict miR-205 binding sites within the PTCH1 3′UTR. Since it would have been inefficient and time-consuming to mutagenize all 11 sites, we decided to narrow down the predicted miR-205 binding sites by constructing three different truncated versions of the PTCH1 3′UTR (Figure 20). Luciferase assays were performed in FLO-1 EAC cells transfected with miR-205 mimic. Simultaneously, we also tested AGS gastric adenocarcinoma cell lines, because this cell line expresses high endogenous levels of both *miR205HG* and miR-205 (data not shown)—thus removing the need to simultaneously transfect a synthetic exogenous miR-205 mimic construct (AGS cells were used instead of NE cells because HEEpiC primary cells are not robust enough to survive luciferase assays). Next, mimic-transfected FLO-1 or AGS cells were transfected with pmiRGLO-full length PTCH1 3′UTR vector or vectors containing pmiRGLO-truncated versions of PTCH1 3′UTR. Relative to pmiRGLO vector-only controls, all three partial 3’UTR inserts showed statistically significant reduction, but full-length 3′UTR insert caused the most significant reduction of luciferase activity in both FLO-1 (*p*-value < 0.00001) and AGS cells (*p*-value < 0.00001) (Figure 21). This result confirms that miR-205, like its host gene *miR205HG*, also targets SHH-signaling, in this case by directly binding to and inhibiting PTCH1. Appendix A shows additional data of NE/BE tissue sample of PTCH1 expression data. 

## 4. Discussion

This project was initiated to study a previously uncharacterized lncRNA, without foreknowledge of directions in which this study might lead. The only directional clue we had at the outset was our finding that this lncRNA was downregulated in the majority of BE and EAC cells and tissues vs. NE cells and tissues. We also knew that this lncRNA hosted a well-studied miRNA. From these limited clues, we designed various functional in vitro and in vivo experiments to elucidate our lncRNA’s biologic functions and its specific mechanisms of action. Our data established that *miR205HG* is downregulated early during neoplastic progression from NE to BE and eventually to EAC; this dysregulation may result in abnormally upregulated SHH transcription and translation, wherein increased availability of this ligand triggers downstream Hh signaling. Our qRT-PCR data validated our initial RNAseq discovery that *miR205HG* was significantly downregulated in both BE and EAC cells vs. NE cells, implying that *miR205HG* dysregulation drives early preneoplastic as well as later neoplastic esophageal transformation. Among 14 additional NE-BE tissue pairs tested, *miR205HG* was downregulated in all, with average fold-change 48.3 (*p*-value < 0.0001); moreover, among 29 NE-EAC tissue pairs tested, *miR205HG* was downregulated in 89.7% of EAC tissues, with average fold-change 68.6 (*p*-value < 0.0001). These impressively high average fold-changes and low *p*-values convinced us of *miR205HG*’s function as an esophageal tumor suppressor gene. Moreover, with its expression in normal cervix, prostate, trachea, and thymus, we reasoned that *miR205HG* could also exert important biological functions in these organs, where its dysregulation could contribute to disease. We speculate that studies in other tissue types, particularly *miR205HG*’s effect on carcinogenesis, could offer benefits to medical research, particularly with discovery of this intriguing lncRNA’s biological function(s) in BE and EAC.

Although miR-205′s specific interaction with *miR205HG* is still unclear, miR-205 also appears to influence Hh signaling by targeting and downregulating PTCH1. It was fascinating to discover that what previously seemed to be a mere miRNA host gene actually possessed independent functional activity, critically altering esophageal cell fate and behavior. Our results thus highlight the broader importance of miRNA host gene lncRNAs and their potential functions. We analyzed NCBI GenBank and MiRBase databases to assess how many lncRNA host genes occur in the human genome. This revealed 158 lncRNAs containing 154 miRNAs. This finding suggests a plethora of lncRNA host genes needing study: such studies will generate further insights into how lncRNAs interact with their host miRNAs in particular, and into functions of all lncRNAs in general. In view of the rapidly burgeoning evidence of the importance of lncRNAs, such insights could prove paradigm-shifting not only in cancer biology and cancer medicine, but in all biology and diseases in general.

Elucidating a particular lncRNA’s biological function(s) is an extremely challenging task. Firstly, the enormous number of existing lncRNAs hinders attempts to categorize their functions, since the majority remains unstudied. Although studies have identified several categories of lncRNA function, additional functions are continuing to be proposed. Secondly, lengths of lncRNAs also make it difficult to predict their functions. Unlike miRNAs, which span only 20–22 nt, lncRNAs are defined as any transcript longer than 200 nt. Given the impact that each short miRNA strand has on its manifold target mRNAs, even short lncRNAs could perform diverse functions. In addition, their length implies that lncRNAs may contain multiple regulatory elements permitting divergent functions to be active at different timepoints and in different contexts or tissues. Thus, considering even just these aspects of lncRNAs, the number of combinations and permutations of imaginable lncRNA functions appears virtually unlimited.

Nonetheless, faced with such a novel lncRNA candidate, we found it helpful to review known functions of other lncRNAs. Current ongoing studies suggest that lncRNAs fulfill a wide variety of regulatory roles at almost every stage of gene expression. *miR-205* was identified as a squamous epithelium-specific biomarker [34]. Taken together, these findings suggest that *miR205HG* dysregulation contributes to development and/or progression of EAC. BE cell line and tissue qRT-PCR results illustrate that *miR205HG* dysregulation may also contribute to neoplastic progression of BE. However, unfortunately, directly injecting BE cells into nude mice is not an option, since these cells do not form xenografts. Nevertheless, we confirmed our novel lncRNA’s tumor-suppressive function in EAC.

We confirmed that several key SHH-signaling genes are upregulated in BE and EAC cells, as expected. More importantly, we discovered that *miR205HG* and SHH expression correlate inversely in matched patient tissue samples. Follow-up experiments revealed that *miR205HG* overexpression inhibits transcription and translation of SHH itself, as well as transcription of SHH pathway genes *PTCH1, SMO* and *GLI1*. While this result is encouraging, it generates additional questions. Firstly, we do not know how *miR205HG* overexpression causes SHH inhibition. It is possible that *miR205HG* interacts directly with the SHH promoter, thereby preventing transcription factors from binding and initiating SHH transcription. Another possibility is that *miR205HG* interacts with Polycomb Repressive Complex 2 (PRC2) and regulates chromatin state in the vicinity of the SHH gene. Future experiments are needed to elucidate *miR205HG*’s specific mechanism(s) of action. Secondly, although SHH and several key SHH-signaling genes are downregulated by *miR205HG*, we have no direct proof that the entire SHH-signaling pathway is inhibited by our lncRNA. We can argue that SHH-signaling is probably inhibited, based on our finding that *miR205HG* overexpression reduces cell proliferation, clonogenicity, and in vivo tumor growth: these results resemble previously reported phenotypes of SHH-signaling inhibition in cancer cells. The effect on in vivo tumor growth was modest, possibly because *miR205HG* influences only early stages of tumorigenesis; however, a Gli reporter construct measuring SHH-signaling activity in *miR205HG*-stably transfected cells may potentially provide stronger direct evidence that *miR205HG* targets SHH and shuts down the entire signaling pathway. Future experiments should also include modulation of in vitro anti-EAC drugs by *miR205HG*, validation of Hh pathway downregulation by *miR205HG* in vivo, exploration of other crucial pathways involved in *miR205HG*-induced tumor suppression, and potential tumor-suppressive synergy between *miR205HG* and its cognate, miR-205. Finally, we were unable to determine whether *miR205HG* levels correlated with poor overall or progression-free survival in EAC patients; this information would be valuable in future studies.

We also explored the cognate transcript miR-205 and its potential synergy with *miR205HG* in SHH-signaling. Luciferase assays established that miR-205 directly binds to and downregulates PTCH1. Since the PTCH1 receptor inhibits SMO, miR-205-induced downregulation of PTCH1 would be expected to result in SMO activation. This data contradicts our initial prediction that *miR205HG* and miR-205 act synergistically, with *miR205HG* inhibiting SHH. One plausible explanation for this conundrum is that miR-205 is involved in a negative feedback loop responding to SHH downregulation by *miR205HG*. An alternative explanation is that these two transcripts do not in fact act synergistically. However, one recent study also reported PTCH1 expression in 58% of BE and 96% of EAC lesions [15]. Notably, data herein establish that PTCH1 is upregulated in BE and EAC cell lines, and forced *miR205HG* overexpression reduces PTCH1 expression in EAC cells. Based on this evidence, *miR205HG* and mir-205 may still act synergistically. Thus, the precise mechanisms of action of the lncRNA *miR205HG* await full elucidation, with attendant complexities and nuances.

## 5. Conclusions

These experiments revealed that *miR205HG* was downregulated in EAC vs. normal esophageal epithelia (NE) as well as in EAC cell lines, and its forced overexpression inhibited EAC cell proliferation and cell cycle progression in vitro. Similarly, overexpression of *miR205HG* inhibited xenograft tumor growth in mice in vivo. Finally, we show that one mechanism of action of *miR205HG* involves the Hh signaling pathway: *miR205HG* and Hh expression levels were inversely correlated in both EAC (r = −0.73) and BE (r = −0.83) tissues, and in vitro studies revealed details of Hh signaling inhibition induced by *miR205HG.* In conclusion, these findings establish that lncRNA *miR205HG* functions as a tumor suppressor in the development of BE and EAC, at least in part through its effect on the Hh signaling pathway.

## Figures and Tables

**Figure 1 cancers-13-01707-f001:**
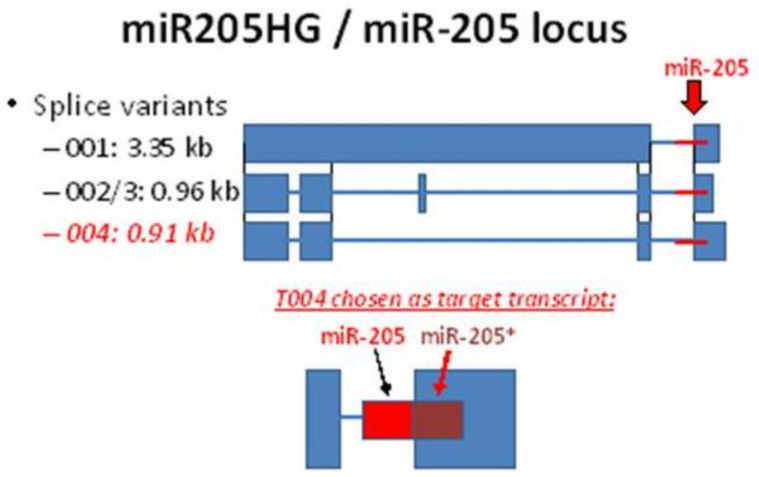
Illustration of *miR205HG* gene locus with three transcript variants and their corresponding sizes (kb).

**Figure 2 cancers-13-01707-f002:**
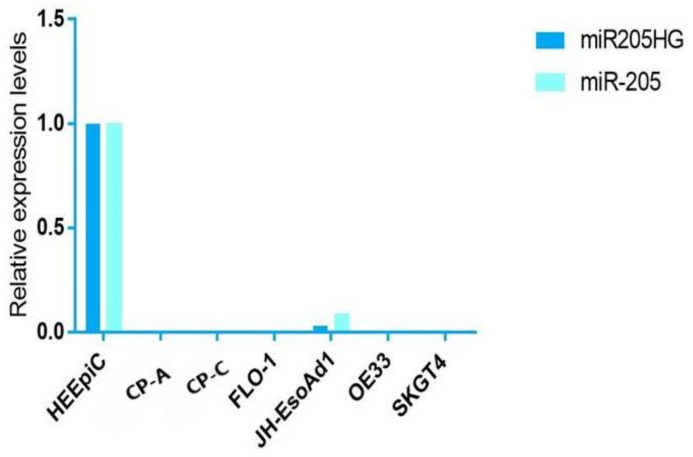
Expression levels of *miR205HG* and miR-205 in BE and EAC cell lines relative to NE (normal esophagus) cells. Expression levels of miR205HG and miR-205 are shown in BE (*CP-C* and *CP-A*) and EAC (OE33, Flo-1, SKGT4, and JH-ea1) cell lines relative to NE (*HEEPiC*-Human Esophageal Epithelial Cells) cells. Both *miR205HG* and miR-205 were neither not detectable or downregulated tenfold or greater in BE and EAC cells.

**Figure 3 cancers-13-01707-f003:**
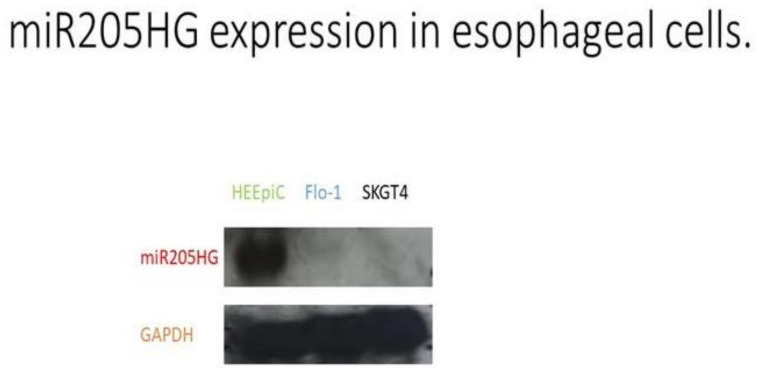
*miR205HG* expression assessed via Northern blotting. glyceraldehyde 3-phosphate dehydrogenase (GAPDH) serves as an internal control. HEEPiC: normal primary esophageal epithelial cells; SKGT4: EAC-derived cell line; Flo-1: EAC-derived cell line (full blots can be found in Appendix A).

**Figure 4 cancers-13-01707-f004:**
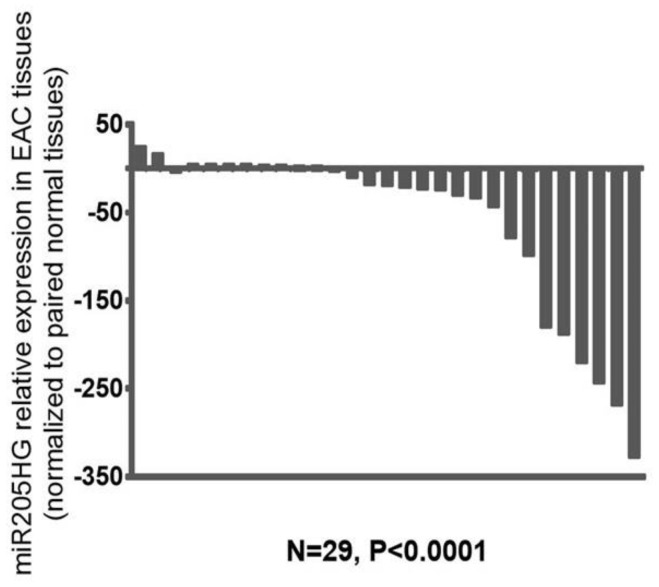
*miR205HG* expression in NE-EAC tissue pairs. Twenty-nine matched NE-EAC tissue pairs were assessed for *miR205HG* expression by qRT-PCR. *miR205HG* expression was downregulated relative to NE in the majority of EACs studied (26/29, average fold-change 68.6, paired *t*-test *p*-value < 0.0001).

**Figure 5 cancers-13-01707-f005:**
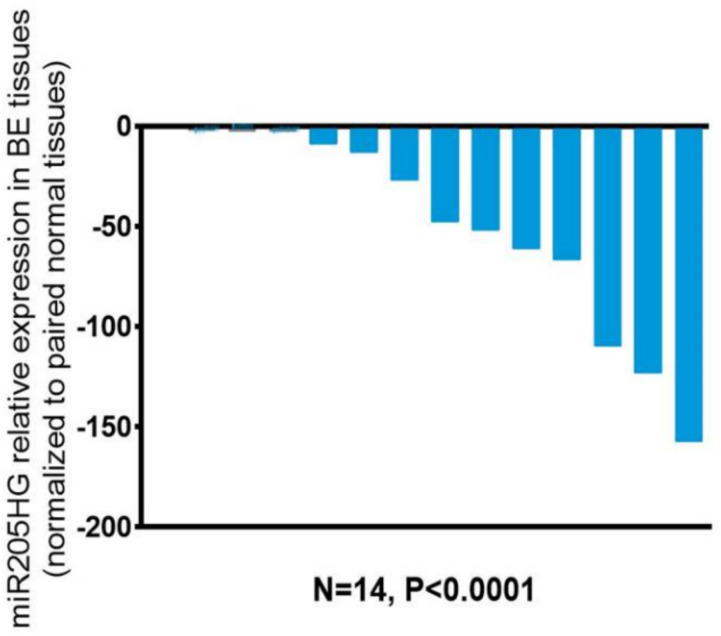
*miR205HG* expression in NE-BE tissue pairs. Fourteen matched NE-BE tissue pairs were assessed for *miR205HG* expression by qRT-PCR. *miR205HG* was downregulated relative to NE in all BE tissues studied (14/14, average fold-change 48.3, paired t-test *p*-value < 0.0001).

**Figure 6 cancers-13-01707-f006:**
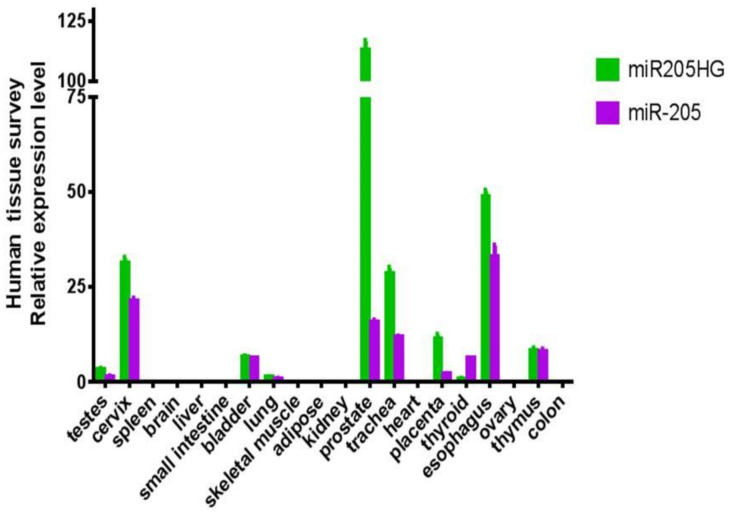
*miR205HG* expression in various normal human tissues. Twenty different normal human tissues were assessed for *miR205HG* expression by qRT-PCR. *miR205HG* was expressed in normal esophagus, cervix, prostate, trachea, and thymus.

**Figure 7 cancers-13-01707-f007:**
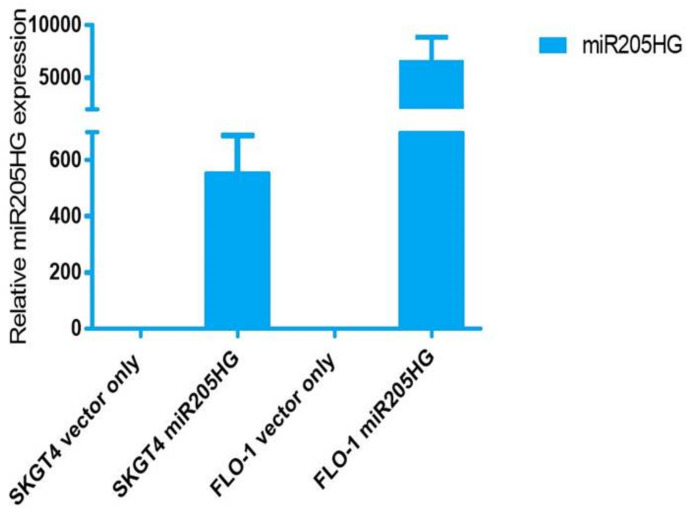
*miR205HG* expression in *miR205HG*-stably transfected clones. SKGT4 and FLO-1 EAC cell lines were *miR205HG*-stably transfected; qRT-PCR confirmed *miR205HG* overexpression in both of these cell lines relative to vector-only transfected cells.

**Figure 8 cancers-13-01707-f008:**
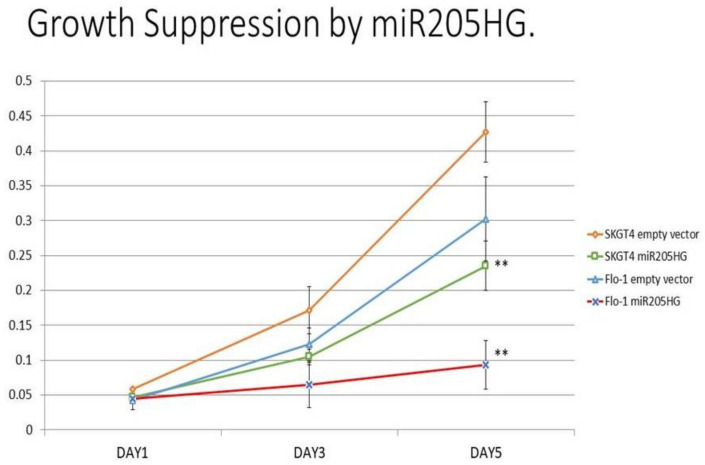
Effect of *miR205HG*-stable transfection on proliferation of EAC cells. *miR205HG* stably transfected clones were studied in water-soluble tetrazolium salt (WST-1) assays to detect the effect of exogenous *miR205HG* overexpression on proliferation of SKGT4 and FLO-1 EAC cell lines. Compared to vector-only transfected negative control, forced overexpression of *miR205HG* decreased cell proliferation on Day 5 by 45% and 69% in SKGT4 and FLO-1 cells, respectively (** *p*-values all < 0.001).

**Figure 9 cancers-13-01707-f009:**
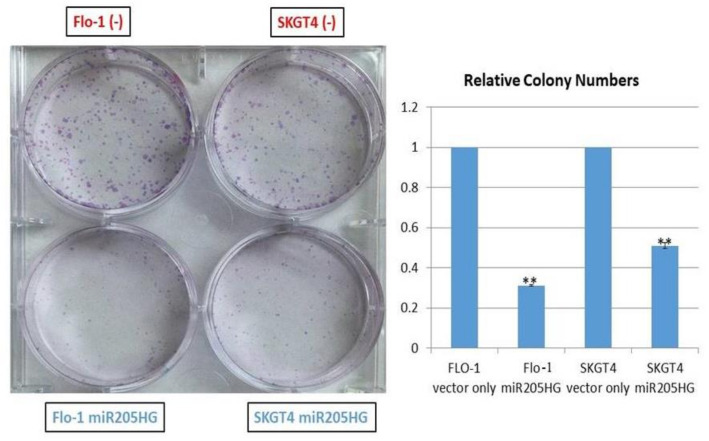
Substantial reduction in colony formation in EAC cells caused by *miR205HG* overexpression. Clonogenic assays revealed that *miR205HG* overexpression induced a substantial and significant reduction in colony formation in SKGT4 (49% reduction; ** *p*-value < 0.02) and FLO-1 (69% reduction; ** *p*-value < 0.01) EAC cell lines.

**Figure 10 cancers-13-01707-f010:**
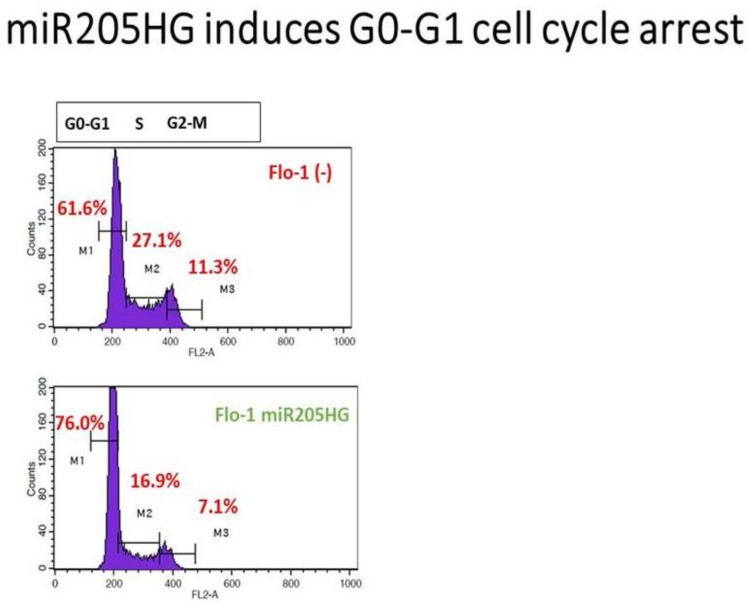
G0/G1 cell cycle arrest induced by *miR205HG* overexpression. Flow cytometric cell cycle assays demonstrated that relative to empty-vector transfected control cells; *miR205HG* overexpression led to an accumulation of cells at G0/G1-phase and a decrease in cells at S-phase.

**Figure 11 cancers-13-01707-f011:**
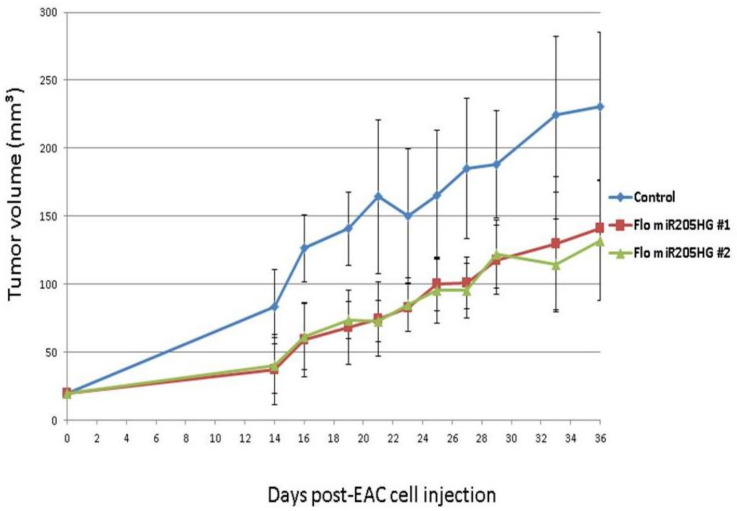
Inhibition of murine tumor growth by *miR205HG*. Fifteen mice were divided into three groups: mice were injected with either stable cells with no insert (control group) or with a miR205HG insert (2 treated groups) and were observed for 36 days. Treated animals began to show significantly smaller tumors on Day 14 (control vs. treatment group 1, *p*-value = 0.003; control vs. treatment group 2, *p*-value = 0.004) onward through Day 36 (control vs. treatment group 1, *p*-value = 0.031; control vs. treatment group 2, *p*-value = 0.017).

**Figure 12 cancers-13-01707-f012:**
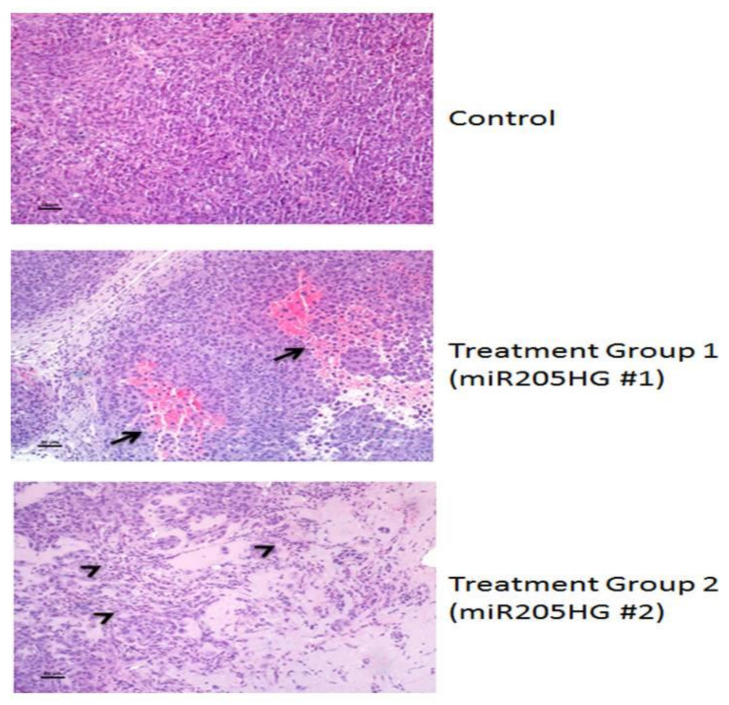
Hemorrhage, necrosis, leukocyte infiltration, and tumor cell death in *miR205HG*-treated murine tumors. Xenografts from vector-treated controls were composed of densely packed sheets of cells with only occasional individual cell necrosis and leukocyte infiltration. In contrast, miR205HG-treated xenografts showed extensive areas of hemorrhage and necrosis (arrows) with leukocyte infiltration (miR205HG #1) or areas of tumor cell death, accumulation of proteinaceous fluid, leukocyte infiltration, and necrotic tumor cells (arrowheads) (miR205HG #2). Scale bars = 50 um.

**Figure 13 cancers-13-01707-f013:**
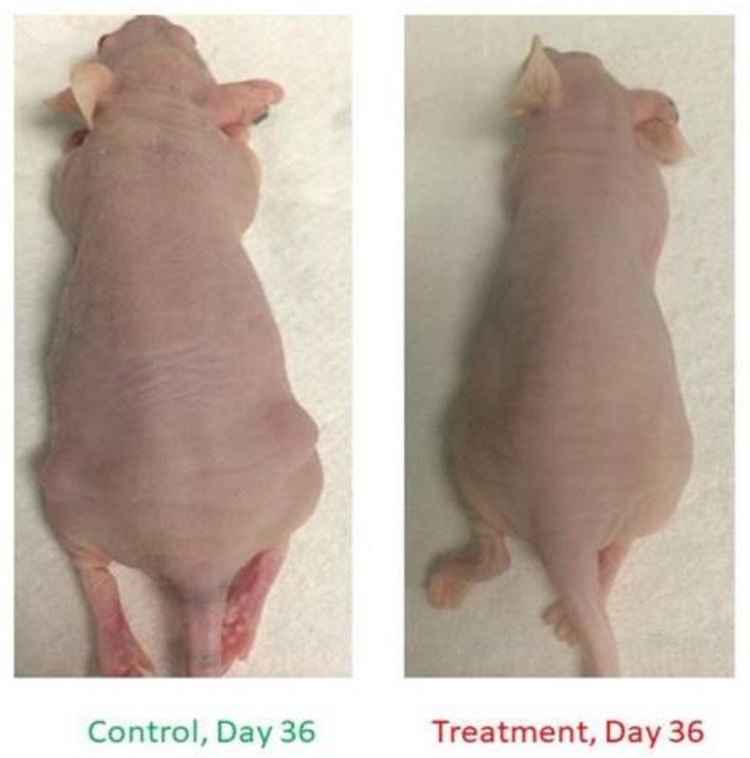
EAC cell xenograft growth inhibition by *miR205HG* in control and treatment mice. Notably, the mean tumor size in treatment group significantly smaller than control group.

**Figure 14 cancers-13-01707-f014:**
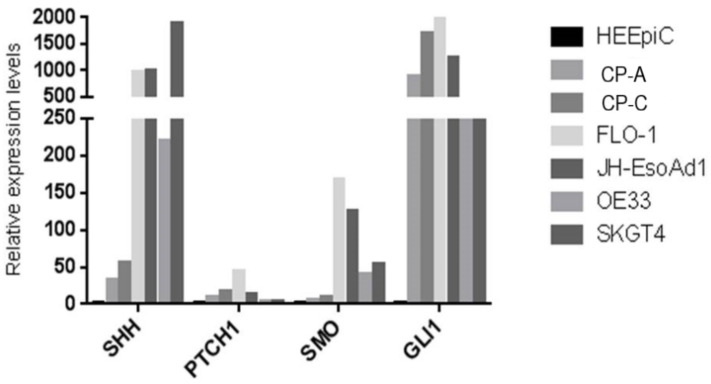
Upregulation of key Sonic Hedgehog (*SHH*) pathway genes *SHH, PTCH1, SMO*, and *GLI1* in both BE and EAC cells. qRT-PCR of BE and EAC cell lines vs. primary NE (HEEpiC) cells revealed that RNA expression levels of the key SHH pathway genes SHH, PTCH1, SMO, and GLI1 were upregulated in both BE (CP-C and CP-A) cell lines and in all four EAC (FLO-1, JH-EsoAd1, OE33, and SKGT4) cell lines studied.

**Figure 15 cancers-13-01707-f015:**
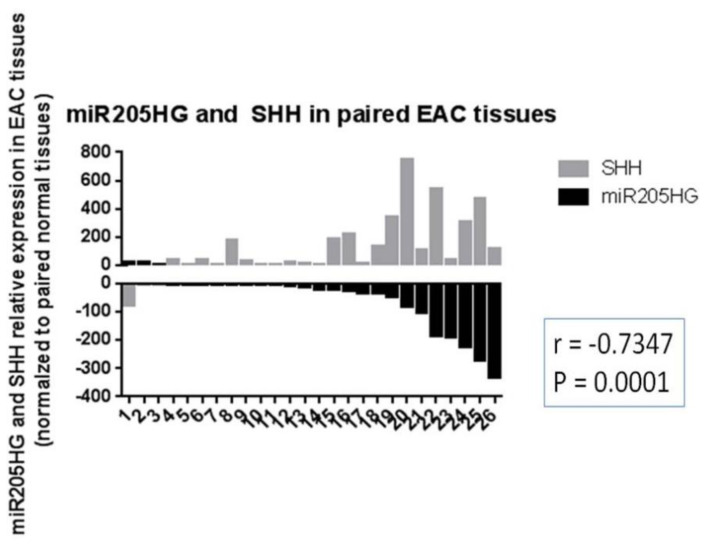
Upregulation of SHH and inverse correlation between *miR205HG* and SHH levels in EAC tissues. Tissues from 26 EAC patients that had already been tested for *miR205HG* expression were also tested for SHH expression by qRT-PCR. SHH expression was significantly upregulated relative to NE in the majority of EACs studied (23/26, average fold-change 154.7, paired *t*-test *p*-value < 0.0001). In addition, *miR205HG* and SHH expression levels were significantly inversely correlated (r = −0.73, *p*-value = 0.0001).

**Figure 16 cancers-13-01707-f016:**
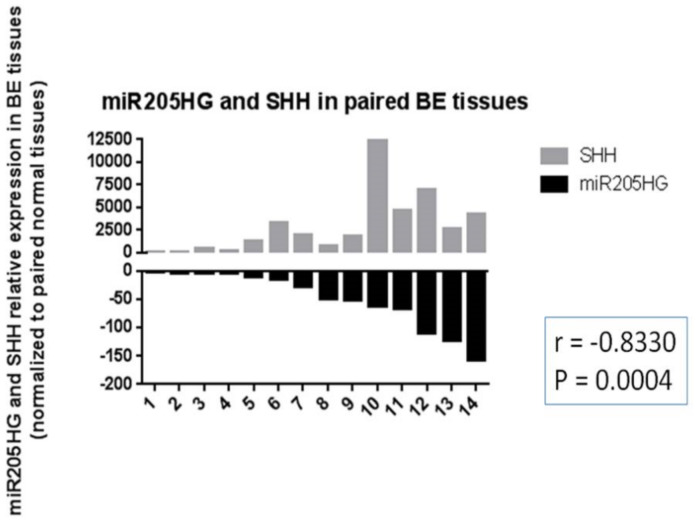
Upregulation of SHH and inverse correlation of miR205HG and SHH expression levels in BE tissues. Tissues from 14 patients with BE were tested by qRT-PCR. SHH was significantly upregulated relative to NE in all BE tissues studied (14/14, average fold-change 2913.0, paired *t*-test *p*-value < 0.0001). In addition, *miR205HG* and SHH expression levels were inversely correlated (r = −0.83, *p*-value = 0.0004).

**Figure 17 cancers-13-01707-f017:**
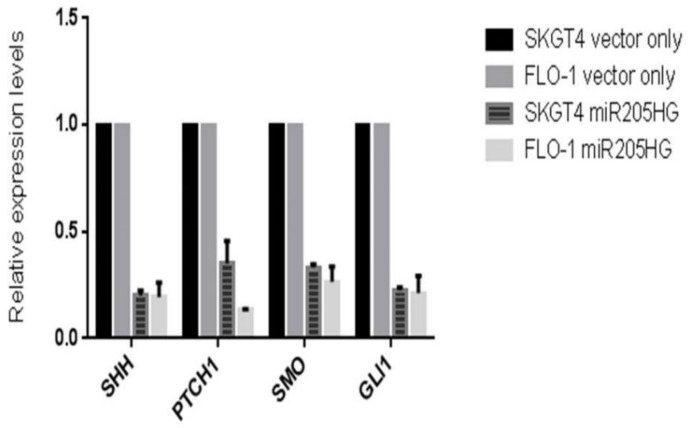
Significant reductions in SHH, PTCH1, SMO, and GLI1 expression levels in miR205HG-stably transfected EAC clones. Relative to vector-only transfected negative control, miR205HG-stably transfected EAC clones exhibited significant reductions in SHH, PTCH1, SMO, and GLI1 expression levels in both SKGT4 and FLO-1 cells (*p*-values for all < 0.05).

**Figure 18 cancers-13-01707-f018:**
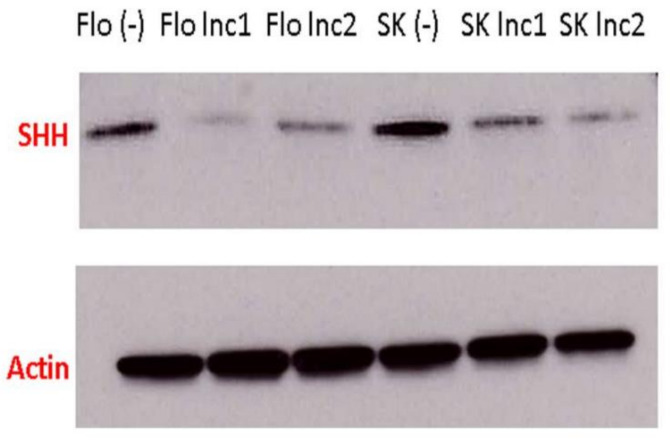
Significant reduction in SHH protein levels in SKGT4 and FLO-1 miR205HG-stably transfected clones. Western blotting demonstrated that SHH protein levels were greatly reduced in SKGT4 and FLO-1 miR205HG-stably transfected clones. Flo (−) and SK (−) denote FLO-1 and SKGT4 cells stably transfected with empty vector only. Flo lnc1 and lnc2 represent two different FLO-1 miR205HG stably transfected clones (the same labeling applies to SKGT4 cells). (full blots can be found in Appendix A).

**Figure 19 cancers-13-01707-f019:**
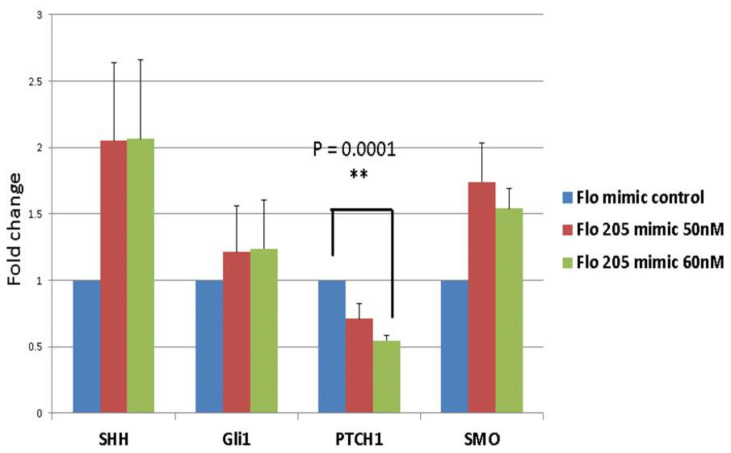
Significant PTCH1 downregulation induced by miR-205 mimic transfection in EAC cells. After transfecting FLO-1 EAC cells with a negative control mimic or a miR-205 mimic, qRT-PCR was performed for four genes (SHH, PTCH1, SMO, GLI1). Only PTCH1 showed a significant (** *p*-value = 0.0001) reduction when transfected with 60 nM of miR-205 mimic.

**Figure 20 cancers-13-01707-f020:**
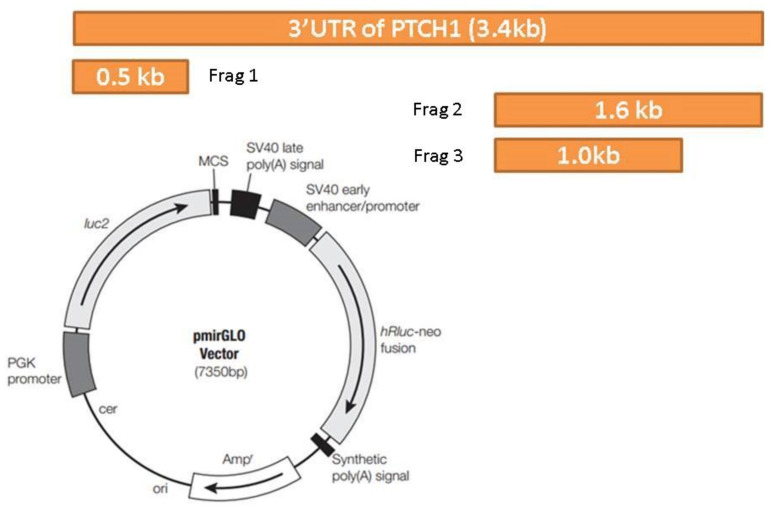
PTCH1 3′ untranslated region (3′UTR) luciferase vector design. The overall PTCH1 3′UTR luciferase vector design is shown above. One full-length PTCH1 3′UTR and three different truncated versions of PTCH1 were constructed and then cloned into vector pmirGLO.

**Figure 21 cancers-13-01707-f021:**
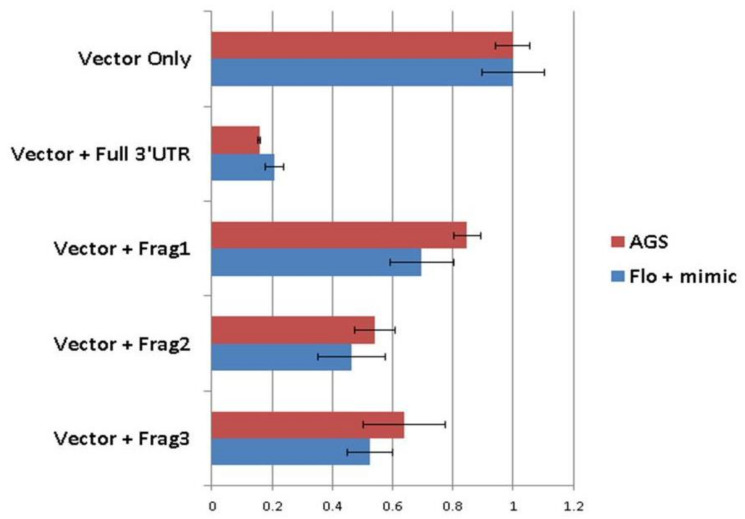
Significant reduction in luciferase activity induced by full-length 3′UTR insert in FLO-1 and AGS cells. Relative to luciferase activity after pmiRGLO vector-only transfection, the full-length 3′UTR insert caused the most significant reduction of luciferase activity in both FLO-1 and AGS cells. Vector + Frag1, 2, and 3 denote three different truncated PTCH1 3′UTR constructs.

**Table 1 cancers-13-01707-t001:** Sequences of forward and reverse primers used for quantitative reverse transcription polymerase chain reaction (qRT-PCR).

Target Gene	Primer Direction	Sequence
*miR205HG*	Forward	GTGCTTTATATAGGAAAGGACCAAC
	Reverse	CCATGCCTCCTGAACTTCACT
*SHH*	Forward	GCTCGGTGAAAGCAGAGAAC
	Reverse	CCAGGAAAGTGAGGAAGTCG
*PTCH1*	Forward	TCCCAGCGCTTTCTACATCT
	Reverse	CTTTGTCGTGGACCCATTCT
*GLI1*	Forward	GTGCAAGTCAAGCCAGAACA
	Reverse	ATAGGGGCCTGACTGGAGAT
*SMO*	Forward	GTTCTCCATCAAGAGCAACCAC
	Reverse	CGATTCTTGATCTCACAGTCAGG

**Table 2 cancers-13-01707-t002:** Top 11 tumor-suppressive lncRNA candidates are sorted by the highest NE/EAC fold change.

Gene Information	Normalized Copy Numbers										Fold Change			
Name	NE1	BE1	EAC1	NE2	BE2	EAC2	NE3	BE3	NE4	BE4	NE/BE	NE/EAC	NE/BE	NE/EAC
*CTA-55I10.1*	1120	11	0	49	0	0	749	157	548	94	101.82	#DIV/0!	4.77	5.83
*AC079305.1*	270	8	60	11	17	3	76	13	52	16	33.75	4.5	5.85	3.25
*NEAT1*	41741	3557	12330	11208	29426	8631	47273	12815	32335	8864	11.73	3.39	3.69	3.65
*RP11-206L10.11*	20	3	7	10	42	22	28	14	38	25	6.67	2.86	2	1.52
*CTC-228N24.3*	99	6	38	16	34	8	34	27	32	10	16.5	2.61	1.26	3.2
*NCRNA00275*	664	28	294	15	38	3	120	44	88	28	23.71	2.26	2.73	3.14
*SNHG12*	240	33	113	13	11	5	49	24	53	18	7.27	2.12	5.04	2.94
*SNHG5*	1294	116	668	19	107	22	218	69	137	36	11.16	1.94	3.16	3.81
*SNHG1*	4665	657	2426	22	37	15	126	29	96	25	7.1	1.92	4.61	3.84
*NBPF1*	128	14	70	22	45	18	60	22	37	22	9.14	1.83	2.73	1.68
*CTC-358I24.1*	70	4	41	10	49	15	54	25	55	26	17.5	1.71	2.16	2.12
*GAS5*	3907	459	2302	30	64	17	219	90	160	66	8.51	1.7	2.43	2.42

RNAseq (RNA sequencing)-generated normalized copy numbers of two matched NE-BE-EAC paired tissues and two matched NE-BE paired tissues are shown above (NE: normal esophagus, BE: Barrett’s esophagus, EAC: esophageal adenocarcinoma). Top 11 tumor-suppressive lncRNA (long non-coding RNA) candidates are sorted by the highest NE/EAC fold change. We prioritized lncRNAs with the highest expression in NE (NE normalized copy number ≥ 10), which were sequentially downregulated during the NE-BE-EAC progression continuum (cut-off fold-change ≥ 1.5). Among 11 lncRNAs shown above, CTA-55I10.1 (subsequently re-named to *miR205HG*) had the highest NE/BE and NE/EAC fold changes.

## Data Availability

The data that support the findings of this study are available from the corresponding author upon reasonable request.

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
