# Peer review of "Novel Long Noncoding RNA miR205HG Functions as an Esophageal Tumor-Suppressive Hedgehog Inhibitor"

_cancers, 2021, doi:10.3390/cancers13071707_

Round 1

Reviewer 1 Report

Comments to the Authors

I find the revision have improved in figures and revised text, however, my major concerns and comments 1,2 and 4 do not be addressed well. Authors just mentions “the transfected cell lines are no longer available” in response 1 and 2; “Due to time constraints and resource limitation at the time of experimentation” in response 4. I understand time constraints of the revision maybe due to journal time limit, and I recommend accepting for publication clarifying or expanding on those questions after revisions, or editor could decide whether the manuscript has sufficiently improved.

This manuscript is a resubmission of an earlier submission. The following is a list of the peer review reports and author responses from that submission.

Round 1

Reviewer 1 Report

Comments to the Authors

The manuscript entitled “Novel Long Non-Coding RNA miR205HG Functions as An Esophageal Tumor-Suppressive Hedgehog Inhibitor” by Dr. Jee Hoon Song. The authors find that lncRNA miR205HG functions as a tumor suppressor in the development of Barrett’s esophagus, precursor to esophageal adenocarcinoma (EAC), at least in part through its effect on the Hedgehog signaling pathway. The authors hypothesized that cognate transcript miR-205 and its potential synergy with miR205HG in Sonic Hedgehog signaling, specifically target on PTCH1-3’UTR. Here, I recommend accepting for publication after revisions to clarify or expand on some of the key findings of the paper.

Major comments:

  1. To compare the expression levels of MIR205HG relative to miR-205, Authors should detect with various of primer pairs spanning the full-length host gene to examine whether MIR205HG transcripts are selectively expressed between esophageal adenocarcinoma and esophageal tumor.
  2. in vitro and vivo animal studies have shown significantly suppressed tumor growth in miR205HG-bearing tumor or miR205HG-transfected cell lines, however, which mechanism of cell growth suppression or cell death involved in miR205HG regulation have to be clarified? Authors could detect apoptosis markers (ex, cleavage PARP, caspases 3/8/9, and so on) or cell cycle regulators, such as cyclins/CDKs, and examine the miR205-5p, miR205-3p, and SHH-related gene and protein expression, specifically PTCH1 expression in miR205HG- and vector- bearing tumors.
  3. It is necessary to connect the relationship among miR205HG, miR205 and PTCH1 in clinical samples that will be enhance the application in the field of translational medicine, diagnosis and prognosis.
  4. Figure 3D was shown that significant reductions in SHH, PTCH1, SMO, and GLI1 mRNA expression levels in miR205HG-stably transfected EAC clones. But how about protein expression of SHH, PTCH1, SMO, and GLI1 in the stable miR205HG-expressed EAC clones compared with vector-expressed cells?
  5. Authors design full length and 3 truncated-region of PTCH1-3’UTR and examine whether miR205 mimic would target to PTCH1-3’UTR. Authors should be clearly to show which sequence and region of PTCH1-3’UTR would be targeted by miR205, firstly. And why only FLO-1 cells are co-transfected pmiRGLO- PTCH1-3’UTR with miR205 mimic in FLO-1 cells, but not AGS cells in Fig. 3H?
  6. In discussion section, Authors might think about and discuss which relationship is direct and important for EAC and esophageal tumor among miR205HG, miR205 and PTCH1 expression.

Minor comments:

  1. Figure 2D, it is better and clear to show three-intendent experiments and bar/column graph.
  2. In Figure 3G, please label which one is Frag1, 2 or 3.
  3. In Figure 3H, if each bar or group have statistically significant?

Author Response

Please see the uploaded Rebuttal Letter. Thank you! 

Reviewer 2 Report

The manuscript by Song et al provides a very detailed and well-structured characterization of role of a novel long non coding RNA miR205HG as a tumor suppressor in Esophageal cancer. The paper is very well written. The role of long non coding RNAs harboring the genes for microRNAs is under studied in general and therefore this study fills this unmet need by elucidating the role of the mir205HG independent of the role of miR205 and 206. With the RNA seq data, this study also opens avenues for future exploration of other lncRNAs in EAC. There is nothing not to like in the manuscript and I really enjoyed reading it.

I have some very minor suggestions/comments.

  1. The choice of the transcript splice variant # 4 among others is not discussed.
  2. Although the authors discuss in detail about the possibility of a synergistic relation between miR205HG and miR205, it was not conclusive. It will be interesting to see if this synergy could be experimentally tested. Similar lncRNAs containing miRNA genes also have the same unresolved issues. For example, in CLL, the lncRNA Dleu2 harbors miRNA15a/16 and the role of Dleu2 is still elusive. A suggestion is that along with construct created to over express the lncRNA devoid of the miRNA coding region. They could have used a full length sequence as well. Thus, an additional control will have provided data for proving synergy if the changes observed with their clone get further affected by having the miRNA sequence included. However, as it stand, the manuscript contains enough and satisfactory information to be accepted without additional experiment. Thus, this suggestion could be used for future work.
  3. The word “is” in line 43 should be removed.
  4. The second half of the sentence on line 45 should be rewritten as “surveyed BE patients” or “surveillance of BE”.
  5. Some places the symbol for micro gram is not inserted properly ( for example: line 160 and 169, a “u” is used for micro). Please check all the symbols and signs.

In summary, this is a very well written, high impact and detailed study. Congratulations to authors for the good work.

Author Response

Please see the uploaded rebuttal letter. Thank you! 
